# A Method for Unsupervised Semi-Quantification of Inmunohistochemical Staining with Beta Divergences

**DOI:** 10.3390/e24040546

**Published:** 2022-04-13

**Authors:** Auxiliadora Sarmiento, Iván Durán-Díaz, Irene Fondón, Mercedes Tomé, Clément Bodineau, Raúl V. Durán

**Affiliations:** 1Departamento de Teoría de la Señal y Comunicaciones, Universidad de Sevilla, Avda. Descubrimientos S/N, 41092 Seville, Spain; iduran@us.es (I.D.-D.); irenef@us.es (I.F.); 2Centro Andaluz de Biología Molecular y Medicina Regenerativa-CABIMER, Consejo Superior de Investigaciones Científicas, Universidad de Sevilla, Universidad Pablo de Olavide, Avda. Américo Vespucio 24, 41092 Seville, Spain; mercedes.tome@cabimer.es (M.T.); cbodineau@bwh.harvard.edu (C.B.); raul.duran@cabimer.es (R.V.D.)

**Keywords:** histopathological images, non-negative matrix factorization, unsupervised stain separation, beta divergence, semi-quantitative scoring, eigendecomposition

## Abstract

In many research laboratories, it is essential to determine the relative expression levels of some proteins of interest in tissue samples. The semi-quantitative scoring of a set of images consists of establishing a scale of scores ranging from zero or one to a maximum number set by the researcher and assigning a score to each image that should represent some predefined characteristic of the IHC staining, such as its intensity. However, manual scoring depends on the judgment of an observer and therefore exposes the assessment to a certain level of bias. In this work, we present a fully automatic and unsupervised method for comparative biomarker quantification in histopathological brightfield images. The method relies on a color separation method that discriminates between two chromogens expressed as brown and blue colors robustly, independent of color variation or biomarker expression level. For this purpose, we have adopted a two-stage stain separation approach in the optical density space. First, a preliminary separation is performed using a deconvolution method in which the color vectors of the stains are determined after an eigendecomposition of the data. Then, we adjust the separation using the non-negative matrix factorization method with beta divergences, initializing the algorithm with the matrices resulting from the previous step. After that, a feature vector of each image based on the intensity of the two chromogens is determined. Finally, the images are annotated using a systematically initialized k-means clustering algorithm with beta divergences. The method clearly defines the initial boundaries of the categories, although some flexibility is added. Experiments for the semi-quantitative scoring of images in five categories have been carried out by comparing the results with the scores of four expert researchers yielding accuracies that range between 76.60% and 94.58%. These results show that the proposed automatic scoring system, which is definable and reproducible, produces consistent results.

## 1. Introduction

Immunohistochemistry (IHC) is a procedure that is widely used and accessible to almost all laboratories for detecting the expression of biological markers in formalin-fixed and paraffin-embedded tissues. Chromogens, which can appear as different colors (brown, blue, red) are localized in fixed tissues to antigens of interest via an antibody–antigen detection system. The appearance of the staining can be followed by observation under bright field microscopy, and standard red–green–blue (RGB) color images can be acquired from it. The main advantage of a chromogenic system is that the quality of the staining is maintained for many years. However, the disadvantage is that the procedure for quantifying the intensity of such staining is not straightforward.

One of the most commonly used chromogens for the visualization of interest proteins is the chromogen 3,3${}^{\prime }$-Diaminobenzidine (DAB), which forms a brown precipitate. The DAB chromogen is not a true absorber of light but a scatterer of light. In addition to absorption, scattering also contributes to the light extinction process and causes a non-linear relation between the absorbance value and the stain amount. In view of this, one can say that DAB does not follow the Beer–Lambert law, which describes the linear relationship between the concentration of a compound and its absorbance, or optical density [1]. Therefore, a precise quantification of the expression level of a protein is not possible through visual examination of a DAB-stained IHC image. However, the use of DAB is widely accepted by the scientific community for the performance of semi-quantitative analysis, especially in comparative studies where images from a trial are ranked according to specific criteria. The number of images within a trial can be variable, but experimental protocols are designed in such a way that in each trial, there will be at least one image with the highest and one with the lowest possible score, which are referred to as control images.

The scoring of immunohistochemically stained tissue samples is important in translational research. However, manual evaluation by an experienced researcher remains a time-consuming and subjective procedure that might lead to intra- and inter-observer variability [2]. Inter-observer variation is the amount of variation between the scores obtained by two or more observers examining the same sample, whereas intra-observer variation is the amount of variation one observer experiences when observing the same sample more than once. In addition, the same sample that is in two different trials may obtain different scores in each trial by the same observer. This is because the criteria for obtaining the highest and lowest scores may differ from one trial to another, which we call inter-session variability. Since it is more objective and reproducible than manual evaluation, semi-quantitative automatic scoring systems are in increasing demand by research laboratories to include histopathology information in biomedical research [3].

There are currently a number of different annotation systems based on different criteria, and in most laboratories, scientists design the scoring system that is best suited to answer a particular scientific question [4]. However, we can see in the literature several systems that are common to many experimental studies. The simplest scoring system for the assessment of IHC staining classifies whole tissue sections as positive or negative, so that the sample is positive when the observed staining is above the background staining level. Other systems are based on estimating the percentage of cells in the tissue or in sub-compartments of the tissue that show positive (or negative) staining without taking into account variations in staining intensity when assigning scores. In addition, scoring systems based on both the staining intensity and percentages of stained cells are employed [5].

Semi-quantitative scoring systems use a tiered approach. In some cases, the signal strength is described by a value within a range on a scale (e.g., grades of 0,1+,2+,3+), whereas in other cases, terms such as “very low or absent”, “low”, “medium”, “high”, or “very high” are used. The number of score categories typically ranges from about three to five. In general, a smaller number of scoring categories tends to reduce the sensitivity of the system, while a larger number tends to reduce reproducibility, as there is less obvious distinction between each of the categories [3]. In any case, it is particularly relevant to pay attention to the specific criteria defining each stain. Note that the criteria may change from one experimental protocol to another, and therefore, supervised techniques developed from one trial based on certain criteria may not work in other trials. Therefore, when developing an automatic system for semi-quantification so that it can be used in a generalized way for the study of any protein of interest, it is recommendable that the methods used to solve the problem be of the unsupervised type.

There are several bioimage analysis platforms that allow implementing scripts for the semi-quantitative scoring of IHC images. However, they often require the adjustment of several software parameters for different images and staining conditions and advanced coding experience, which could be a tedious and time-consuming task for such a workflow to be applicable. Two of these freely distributed platforms are QuPath [6] and IHC Profiler [7], which is a plugging of the free software ImageJ Fiji. Researchers that perform image analysis with these platforms, as for example in [8,9], often do not provide enough detail to reproduce the step-by-step analysis protocols in their publications, although the protocols can be quite tedious and time consuming for researchers, as they are not fully automatic. A practical guide to semi-quantification with the IHC profiler is detailed in [10], in which the necessary researcher intervention in the procedure described above can be verified, as for example for the selection of appropriate cut-off thresholds for protein detection.

In this work, we present a fully automatic and unsupervised method for comparative biomarker quantification that relies on a color separation method that discriminates between DAB (brown) and H (blue) staining robustly, independent of color variation, or biomarker expression level. To the best of our knowledge, this is the first end-to-end semi-quantification system that takes a set of IHC images as the input and directly outputs a score to each of them without the intervention of the researcher beyond specifying which are the control images.The proposed method employs: a two-step stain separation procedure based on eigendecomposition (ED) and NMF approaches, features based on the stain concentration matrix and hue saturation of color vectors estimated, and a clustering k-means approach for predicting the semi-quantitative scores. We use the family of beta divergences as the dissimilarity measure in both NMF and k-means procedures due to its proven robustness against outliers. The developed method is designed to predict the image scores as an expert researcher would. Experimental results show that our method is able to predict the image scores on a 5-point scale with very high and statistically significant correlation with experienced researchers’ scores.

## 2. Related Work

In many IHC experimental procedures, the biological samples are stained with two stains, traditionally DAB and hematoxylin (H). When double staining is performed, chromogens are selected to provide maximum color contrast when observed. In the above case, the corresponding colors are brown and blue, respectively. However, color is a perception generated in the brain of a person that cannot be considered as isolated values, as they are surrounded by other colors that can alter their perception. There are several visual and cognitive traps described in the literature that can affect manual image scoring and that can be avoided by using automatic image analysis systems. For instance, colors can be enhanced or muted depending on the colors that surround them. A detailed description of these kinds of effects affecting IHC images can be found in [5]. For this reason, many annotation systems separate the color planes to allow researchers to visualize each stain separately. In addition, stain separation makes it possible to calculate features related to the intensity of each staining independently.

Although it is well established that the Beer–Lambert law only applies rigorously when only absorbance contributes significantly to attenuation of the light, optical density (OD) space is usually used as a rapid proxy model to assess the stain separation of IHC images, even considering the presence of certain scattering. The assumption that the intensity in the OD space is roughly linear with the amount of each stain absorbed by a tissue sample allows the use of many well-known signal processing techniques to overcome the problem of staining separation. In addition, the staining planes obtained with the separation allow performing merely comparative studies, although they do not allow determining the precise amount of stain concentration in the tissue.

Let $I\in R^{3\times N}$ be the matrix of RGB intensities where *N* represents the number of pixels of the image, and let $I_{0}$ be the illuminating light intensity on the sample (usually 255 for an 8-bit image). If each color plane is stacked in a column vector $i_{c}={\left[i_{1,c},\cdots ,i_{N,c}\right]}^{T}$, c∈{R,G,B}, the OD of a image channel is defined as
(1)$$y_{c}=-\log_{10}\left(\frac{i_{c}}{I_{0}}\right),$$
where the values for $y_{c}$ are computed element-wise. The observed OD image $Y={\left[y_{R},y_{G},y_{B}\right]}^{T}$ can be decomposed as
(2)Y=WH
where $W\in R^{3\times n_{s}}$ is the color vector matrix, $H\in R^{n_{s}\times N}$ contains the stain concentrations or stain density maps, and $n_{s}$ represents the number of stains considered. Note that each column of W contains the color composition of the *s*-th stain, whereas each row of H contains the whole contribution of the *s*-th stain to the image.

There are several methods to approach staining separation, which mainly fall into one of the following two families: methods based on color deconvolution and methods based on blind source separation. Color deconvolution (CD) algorithms are based on the Beer–Lambert law and require either a priori knowledge or estimation of color vectors of each specific stain, which are to un-mix the stain concentrations on a per pixel basis. The color vector matrix W can be estimated empirically through fixed RGB triplets for the two stains [11], by singular value decomposition approaches [12,13,14], by applying Bayesian modeling and inference [15], or by using k-means clustering [16,17]. Even in a well-controlled experimental protocol, IHC-stained slides may suffer inter-batch staining variations, and therefore, the color vectors have to be estimated for each image independently, which is time consuming for researchers. Moreover, one major drawback of these methods is that good results are not obtained when one of the stains is not sufficiently represented in the image, this being one of the main characteristics of comparative studies. Therefore, the use of deconvolution techniques for semi-quantitative studies often requires the intervention of the researcher to determine successfully the base colors.

Several blind source separation algorithms have been also employed for stain separation such as Independent Component Analysis (ICA) [18], Non-Negative Least Squares (NNLS) [19], and Non-Negative Matrix Factorization (NMF) using the Euclidean distance in the cost function as well as regularization and sparsity constraints [20,21]. For example, the method proposed in [21] called Sparse Non-Negative Matrix Factorization (SNMF) added a sparseness regularization term to the stain concentration matrix H. The inclusion of these regularization terms is based on the assumption that each stain is fixed only to specific tissue structures, forcing most pixels to become reactive to only one type of stain. However, this assumption may not hold true in many experiments, as it will depend on the protein of interest under study. Moreover, the use of regularization techniques may not work as expected in images where the intensity of the chromogen is low. The major drawback of methods based on blind source separation algorithms is that the behavior in the IHC context is strongly dependent on their initialization [22]. Stain decomposition may converge to any local minima without a proper initialization, leading to wrong estimation if the initial color bases are not close enough to the true stain spectra. In spite of this, most authors opt for a random type of initialization, such as the RGB optical density of randomly selected pixels corresponding to two columns of W of the histological image proposed in [21]. Clearly, random initialization does not guarantee that the solution found is adequate, and therefore, some authors use more complex initialization methods as in [23], where a saturation-weighted statistical method is developed.

The method proposed in [24] creates binary masks based on color histograms. These masks are used to mask the R, G, and B channels to obtain color images that represent the segmented blue and brown stains. However, staining planes obtained by segmentation have two main drawbacks. On the one hand, it is very difficult to adjust the masks to different background noises, and the resulting images often show quite clearly areas where background noise is still present. On the other hand, segmentation does not separate staining in regions where co-localization is present. In most cases, regions with co-localization are assigned to only one of the stains, which leads to subsequent errors in image interpretation.

Finally, frameworks based on deep learning approaches [25,26] have been also developed. Two important limitations of deep learning models are that they require a large amount of images for model training, and that the model should be applicable to stainings from any type of protein and tissue and generalizable to datasets generated by different laboratories. In [25], the team developed a multi-label classification deep model for IHC images of human testis, with a dataset generated as a part of the Human Protein Atlas (HPA) project [27,28] for which a manual annotation had to be performed. The results obtained are promising although limited to that specific type of tissue. In [26], a Convolutional Neural Network (CNN) was employed to annotate the IHC images of neurons. The system first recognizes and crops the stained neurons with the CNN; then, it obtains the cytoplasmatic DAB signal by deconvolution, and finally, it measures the average intensity of the cytoplasm of the neurons. The overall inter-rater agreement obtained was poor.

Once the DAB staining plane has been obtained, the next step consists of calculating one or more features for quantifying biomarker expression that allow image scoring. For this purpose, several scoring methodologies make use of a pixelwise strategy that does not require the detection and delineation of individual cells and their sub-cellular compartments. The simplest method, sometimes referred to as positive pixel counting, consists of measuring the relative proportion of brown pixels above and below a fixed threshold to be set by the researcher. In this way, the pixels belonging to background staining are not counted. However, the background threshold can differ from one image to another, which makes it necessary to adjust the decision threshold in each image. Two more sophisticated methods are the average threshold method (ATM) [29] and the pixel-wise H-score [30]. The ATM score is a weighted average of all the pixels in the DAB channel given by
(3)$$\mathrm{ATM}\mathrm{score}=\frac{1}{I_{0}}\sum_{k=1}^{I_{0}}PS\left(k\right)$$
where PS(k) denotes the proportion of pixels with intensity greater that or equal to *k* and $I_{0}=255$ assuming 8-bit resolution. In [29], the DAB channel was calculated as the blue complement of the original RGB image. The determination of the ATM score implies that the DAB channel must be converted to grayscale.

The pixel-wise H-score is based on both the intensity and the proportion of the biomarker of interest in the IHC image. For computing the pix H-score, pixels in the DAB staining plane have to be classified as high, medium, and low, and positive pixels in the H channel have to be detected. The thresholds for DAB and H staining planes have to be selected specifically for each biomarker by the researcher. Then, the pix H-score is given by
(4)$$\mathrm{pix}\;\mathrm{H-score}=100\frac{3H_{P}+2M_{P}+L_{P}}{H_{P}+M_{P}+L_{P}+N_{P}}$$
where $H_{P},M_{P},L_{P}$, and $N_{P}$ denote the area of DAB high, DAB medium, DAB low, and H-positive pixels, respectively. In [30], the DAB and H staining planes were obtained by deconvolution.

## 3. Methods

The proposed method first converts the RGB images into optical density using the Beer–Lambert law, as shown in Equation (Equation 1). After this, the following three stages are carried out: stain separation, feature extraction, and prediction of the scores. The separation of hematoxylin-stained nuclei from DAB-positive structures allows the automated extraction of appropriate features for each stain independently. This stage is crucial to achieve robust and accurate algorithms, with any inconsistency at this stage greatly affecting the estimation of the stain concentration. After that, three features related with the intensity of DAB and H chromogens are calculated. Finally, the scores for all the images are predicted through an unsupervised clustering strategy. A precise description of the proposed method is given below.

### 3.1. Stain Separation

Images obtained in the same laboratory can be subject to quite large differences in the true colors and intensity expressed by the stains. Thus, the staining separation method must estimate the color vectors of the stains independently for each image and must be able to correctly separate high and low-intensity stains. A common practice for experimental protocols in semi-quantitative studies is to include control images with the highest and lowest score. Our method assumes the existence of such control images so that the investigator must indicate what these images are. We denote the OD of the control images with the maximum and minimum score as $Y_{5+}$ and $Y_{1+}$, respectively.

In images where there is a high expression of the protein of interest, and therefore, the brown color is expressed with high intensity, traditional unsupervised separation methods, such as those based on CD or NMF, usually yield good results. However, in images where the protein of interest has a low expression, traditional methods may fail, and it is often necessary to determine manually the stain color vectors. To solve this problem, our separation method consists of two steps. In a first preliminary separation step, the expert has to choose a control image with the highest score from the set of images. On this control image, a separation based on ED is performed to obtain the initial staining color vectors. Then, the staining concentration matrix of each image is calculated by a traditional deconvolution method. In the final separation step, an NMF algorithm is initialized with the initial matrices calculated in the first step. An NMF algorithm with beta divergences has been chosen, because several authors have stated the usefulness of beta divergences as a robust similarity measure against outliers [31]. IHC images may contain artifacts generated during tissue processing, hence the need for robust similarity measures. This second stage allows a better adjustment of the solution by automatically updating the color and the concentration of the separate stains. A precise description of the proposed stain separation method is given below.

#### 3.1.1. Preliminary Separation Step

We obtain the color vector matrix of $Y_{5+}$, $W_{init}$, through an eigendecomposition procedure. This color vector matrix is employed as the initial color vector matrix for all the images in the trial, $W_{init}^{m}=W_{init},\forall m=1\cdots M$, in which *M* is the total number of images in the trial.

Since $Y_{5+}$ is generated, predominantly, by the combination of two vectors (those produced by DAB and hematoxylin), its scattering mainly lies in the plane spanned by both vectors (the columns of $W_{init}$, its color vector matrix). It is well known that the ED of the autocorrelation matrix of $Y_{5+}$, $R_{Y_{5+}}=Y_{5+}Y_{5+}^{T}/N$, provides a basis for the above-mentioned plane [32]. This basis is given by the eigenvectors associated to the 2 largest eigenvalues of $R_{Y_{5+}}$. Therefore, the first step of the initialization is to compute the ED of $R_{Y_{5+}}$, obtaining its eigenvector matrix, $Q=[q_{1},q_{2},q_{3}]$, where the first column corresponds to the largest eigenvalue and the third column corresponds to the lowest one. The second step is the computation of the 2 first principal components (i.e., the coordinates of $Y_{5+}$ in the first 2 columns of Q),
(5)$$Z={\left[q_{1},q_{2}\right]}^{T}Y_{5+}=\underset{\left[v_{1},v_{2}\right]}{\underbrace{{\left[q_{1},q_{2}\right]}^{T}W_{init}}}H.$$

The 2D vectors $v_{1}$ and $v_{2}$ are the coordinates of the columns of $W_{init}$ in $\left[q_{1},q_{2}\right]$. Therefore, $W_{init}=\left[q_{1},q_{2}\right]\cdot \left[v_{1},v_{2}\right]$. The third step is to find good estimates of $v_{1}$ and $v_{2}$. Since nuclei are mainly H-stained and the cytoplasms is DAB-stained, there will be a lot of pixels with high value of only one of the stains (one of the rows of H), although there will be also a lot of pixels with high values of both stains. So, defining z(t) as the *t*-th column of Z and ϕ(t)=∠(z(t)) as the angle given by both coordinates of z(t), the search of the estimates of $v_{1}$ and $v_{2}$ is made by means of the search of 2 angles ($\phi_{1}$ and $\phi_{2}$) with a large concentration of high levels of ∥z(t)∥. For this purpose, the range of values of ϕ(t) is divided into 1000 intervals. For the *k*-th interval, the maximum of ∥z(t)∥ is taken as ν(k). A smoothing filter (local mean for 20 samples) is applied to ν(t). Since the vectors $v_{1}$ and $v_{2}$ are near the edges of the 2D scattering, the angles $\phi_{1}$ and $\phi_{2}$ are estimated by searching for the first and last peaks of the filtered version of ν(k). If for some control image, both peaks cannot be estimated, $\phi_{1}$ and $\phi_{2}$ are taken at 18% and 55% of the whole range, respectively.

Once the initial color vector matrix has been estimated, the initial stain concentration matrix for each image *m* in the trial is obtained through color deconvolution [11]
(6)$$H_{init}^{m}=W_{init}^{+}Y,\;\;Y\geq 0$$
where $W_{init}^{+}={\left(W_{init}^{T}W_{init}\right)}^{-1}W_{init}^{T}$ is the Moore–Penrose pseudo-inverse matrix of the non-square matrix $W_{init}$ selected. In addition, all the negative elements of matrix $H_{init}^{m}$ are set to a very small value, $10^{-16}$, since the molar concentration of a specific stain must be a positive value.

#### 3.1.2. Final Separation Step

In the final separation step, the decomposition of each image in the trial is adjusted using NMF with beta divergences initializing the algorithm with the matrices resulting from preliminary separation. Since the variation of the color vector matrix is expected to be small among all the images on a trial, the initialization matrices obtained in the previous step are expected to be close to a desired solution. Therefore, a small number of iterations of the NMF algorithm are needed, thus reducing the computing time.

The beta divergences are a family of divergences parameterized by a single shape parameter β [33]. Given two non-negative matrices $P\in R_{+}^{I\times T}$ and $Q\in R_{+}^{I\times T}$ with entries $p_{it}={\left[P\right]}_{it}$ and $q_{it}={\left[Q\right]}_{it}$, using the notation given by [34], the beta divergences can be expressed as
(7)$$D_{B}^{\beta }\left(P∥Q\right)=\sum_{it}d_{B}^{\;\beta }\left(p_{it},q_{it}\right)$$
where $d_{B}^{\beta }\left(p_{it},q_{it}\right)$ is defined as
(8)$$d_{B}^{\beta }\left(p_{it},q_{it}\right)=\left\{\begin{array}{ccc} -\frac{1}{\beta }\left(p_{it}q_{it}^{\beta }-\frac{1}{1+\beta }p_{it}^{1+\beta }-\frac{\beta }{1+\beta }q_{it}^{1+\beta }\right), & \mathrm{for} & \beta ,1+\beta \neq 0\\ \frac{1}{2}{\left(p_{it}-q_{it}\right)}^{2}, & \mathrm{for} & \beta =1\\ p_{it}\ln\frac{p_{it}}{q_{it}}-p_{it}+q_{it}, & \mathrm{for} & \beta =0\\ \ln\frac{q_{it}}{p_{it}}+\left(\frac{p_{it}}{q_{it}}\right)-1, & \mathrm{for} & \beta =-1 \end{array}\right.$$

The beta divergences include several well-known divergence measures as special cases of this parameter and thus allowing smooth interpolation between many known divergences. In particular, when β=0, the beta divergence takes the form of the standard Kullback–Leibler (KL) divergence, whereas when β=−1, it reduces to the Itakura–Saito (IS) divergence. In addition, the beta divergence corresponds to the standard squared Euclidean distance for β=1.

The appropriate value for the β parameter is related with the distribution of the data and the statistics of the noise assumed on the data. For example, using the Euclidean distance corresponds to the maximum likelihood estimator for i.i.d. Gaussian noise. Similarly, the KL and the IS divergences correspond to a Poisson distribution and multiplicative Gamma noise, respectively [35]. Although the Euclidean distance is the most widely used to tackle the stain separation problem, other dissimilarities measures should be explored. In fact, beta divergences have been proven to be robust to outliers for some values of the tuning parameter β being more suitable than other metrics for some specific applications [36]. For the stain separation, we will assume that $P=[p_{it}]$ is the observed OD image Y, whereas $Q=[p_{it}]=WH$ is the decomposition. The objective of the NMF is to solve the minimization problem subject to the non-negativity constraints defined by
(9)$$\min_{W,H}D_{B}^{\beta }\left(Y∥WH\right),\;\;s.t.W\geq 0,H\geq 0$$

A multiplicative NMF algorithm equipped with the beta divergence as the approximation measure was developed in [37]. The corresponding multiplicative update rules are given by the following equations
(10)$$\begin{array}{c} h_{jt}\leftarrow h_{jt}\left(\sum_{i=1}^{I}\frac{w_{ij}q_{it}^{\beta -1}p_{it}}{\sum_{i=1}^{I}w_{ij}q_{it}^{\beta }}\right) \end{array}$$
(11)$$\begin{array}{c} w_{ij}\leftarrow w_{ij}\left(\sum_{t=1}^{T}\frac{h_{jt}q_{it}^{\beta -1}p_{it}}{\sum_{t=1}^{T}h_{jt}q_{it}^{\beta }}\right) \end{array}$$

In practice, at each iteration of the algorithm, the columns of the matrix W are normalized to the unit $\ell_{p}$-norm (typically, *p* = 1). This normalization is actually a normalization of the chromogenic color vectors. Thus, the staining concentration matrices of two images can be compared even if the color vector matrices of the images are not exactly the same. The difference in the color vector matrices between two images is due to differences in the hue and saturation of the true colors, which is a common phenomenon in IHC comparative studies.

It is well established that the monotonic descent in the beta divergence with the update formulas of Equations (Equation 10) and (Equation 11) is guaranteed for β=(0,1). However, convergence to a local minimum can also be achieved for β heats outside this range; see [34] for more information. In our experiments, we have set the value of the β parameter of the divergence to 0.5.

Since convergence to a global minimum is not guaranteed, only convergence to a local minimum, it is very important to pay attention to the initialization of the NMF algorithm. One of the most commonly used techniques when no prior information is available is to perform several tests with random matrices. Then, the final decomposition corresponds to the matrices obtained in the test with the minimum cost function value. Such an initialization means that the initial point of the iterative process might be close to a local minimum, and therefore, the separation can converge to unsuitable solutions, with stain color vectors very different from those expected. Hence, our method initializes the NMF algorithm with the decomposition $W_{init}^{+}H_{init}^{m}$ obtained in the preliminary separation for each image $Y^{m}$.

Once the final decomposition is obtained, the matrices are arranged so that the first column of $W^{m}$ and the first row of $H^{m}$ correspond to the hematoxylin chromogen, and the second column of $W^{m}$ and the second row of $H^{m}$ correspond to the DAB chromogen. For that, we compare the average value of the magenta channel of the CMYK color model of the separated stains.

### 3.2. Feature Extraction

Once the DAB and H plane have been obtained, the next step is to calculate one or more features whose value allows the images to be properly classified in a similar way to what an expert researcher would do. We have calculated a feature vector $f^{m}=\left[f_{1}^{m},f_{2}^{m},f_{3}^{m}\right]$ for each image $Y^{m}$ that is composed of three features related to the amount of DAB concentration encoded in the matrix $H^{m}$ of the decomposition. The first one, $f_{1}^{m}$, is the 1-norm of the DAB staining plane normalized to the image size
(12)$$f_{1}^{m}=\frac{1}{N}\sum_{i=1}^{N}H^{m}\left(2,i\right),$$
in which *N* is the number of pixels in the image. This feature, unlike others such as the pix H-score and the positive pixel counting, does not require the setting of any parameters or decision thresholds.

Features $f_{2}^{m}$ and $f_{3}^{m}$ are related with the normalization procedure presented in [21]. This normalization method uses normalization factors for each stain in order to preserve the positive structures encoded in the stain concentration matrix, while color appearance normalization is performed. We employ the control image $Y_{5+}$ as a reference image so that the normalization factors are calculated as follows
(13)$$N_{j}^{m}=\frac{H{\left(j,:\right)}^{RM,5+}}{H{\left(j,:\right)}^{RM,m}},\;\;j=1,2$$
where the superscript RM denotes the robust pseudo maximum of each row vector at 99%. These normalization factors can be interpreted as a relative intensity measure between the control image $Y_{5+}$ and the image $Y^{m}$. Then, the second and third features are computed as
(14)$$f_{2}^{m}=N_{2}^{m}$$
(15)$$f_{3}^{m}=\frac{N_{2}^{m}}{N_{1}^{m}}$$

Note that the values of features $f_{2}^{m}$ and $f_{3}^{m}$ do not depend on the color bases determined at the stain separation stage.

### 3.3. Prediction of the Scores

The images are scored using a k-means clustering algorithm equipped with beta divergences [38] for β=−0.5 and initializing the initial centroids in a methodical way using the references images $Y_{5+}$ and $Y_{1+}$. We have not considered it suitable to set fixed boundaries for the classification because we have not found clear boundaries between the different classes in the dataset. Instead, we have opted for a clustering algorithm that allows some flexibility in determining the boundaries of each class. This added flexibility achieves consistent emulation of expert annotations. We have reduced the number of iterations to a maximum of 3 iterations to avoid that the final centroids are too different from the initial centroids, since the distribution of the data is a priori unknown, and some of the classes may be under-represented.

The initial centroids of each cluster have been calculated using a systematic approach. The values for the initial centroid of clusters “5+” and “1+” are the features of the control images $Y_{5+}$ and $Y_{1+}$ selected by the researcher in each trial. The first feature of the initial centroids for the groups scores as “2+”, “3+”, and “4+” are calculated as a percentage of the difference between the first feature of the two control images,
(16)$$f_{1}^{c_{k}}=f_{1}^{1+}+\frac{p_{k}}{100}\left(f_{1}^{5+}-f_{1}^{1+}\right),\;\;k=2,3,4$$
where $c_{k}$ denotes the *k*-centroid and $p_{k}={\left[p\right]}_{k}$ represents the percentage associated with the *k*-centroid. These percentage can be fixed by the researchers, although in our experiments, we have used values of 15%, 40%, and 70% to initialize the groups scored as “2+”, “3+”, and “4+”, respectively, that is, p=[15,40,70].

Although in principle, one might expect a linear relationship between feature $f_{1}^{m}$ and features $f_{2}^{m}$ and $f_{3}^{m}$, our experiments have not revealed such a relationship but rather an exponential type. Thus, for the determination of the second and third features of the initial centroids, we have fitted the feature vectors to a two-term exponential model $y_{t}\left(x\right)=a\exp\left(bx\right)+c\exp\left(dx\right)$ with $x=f_{1}$ and t=2,3. Then, we have predicted the values for $f_{2}^{c_{k}}$ and $f_{3}^{c_{k}}$ as follows
(17)$$f_{t}^{c_{k}}=y_{t}\left(f_{1}^{c_{k}}\right),\;\;k=2,3,4\;\;t=2,3$$

## 4. Results and Discussion

### 4.1. Data Description

Our dataset consisted of 94 images taken from stained xenograph tumors, using HCT116 cells implanted in mice treated with either vehicle or with a cell-permeable *α*-ketoglutarate derivative, dimethyl*α*-KG (DMKG), followed by treatment with the mTORC1 inhibitor temsirolimus (TEM) or with metformin (MET). Samples were processed as described in [39]. All procedures were approved by the corresponding institutional organizations (APAFIS# 10090 2017052409402562 v2). Omission of the primary antibody in the immunostaining procedure was used as a negative control. Images were acquired in the TIFF format with a Leica DM6000B microscope using ×20 or ×40 objective lenses and a Leica DFC500 digital camera. All images have been independently annotated by four expert observers. Figure 1 shows examples of different IHC tissue images scored from “1+” to “5+” in the same way by all the observers.

The agreement among observers measured as the percentage of images that have been annotated with the same score by all experts is 70.21%. Figure 2 shows some of the images where there is a discrepancy between the annotations of the observers.

### 4.2. Stain Separation Results

The two-stage stain separation method achieves satisfactory DAB and H planes in both high-intensity and low-intensity images. The hue saturation of the blue and brown colors may differ noticeably between some of the images in the database. Therefore, as mentioned above, it is necessary to adjust the color bases of each image with respect to the bases obtained in the control image. Figure 3 shows the stain separation results of the “5+” control image obtained through eigendecomposition, whereas Figure 4 and Figure 5 show the preliminary and final results of staining separation in high and low-intensity level images, respectively. The better adjustment of the brown and blue colors in the final separation results compared to the preliminary separation results can be seen.

### 4.3. Prediction of the Scores Results

We have investigated the suitability of several pixel-wise features as discriminating features to be able to differentiate between the different categories of images. In particular, we have calculated the ATM score, the pix H-score, and the 1-norm of the DAB stain concentration vector obtained in the stain separation stage of all the images in the dataset. The ATM score has been calculated using the RGB image of the DAB plane. For computing the pix H-score, we have manually adjusted the thresholds to {2.5,0.7} values for the DAB plane and {0.7} for the H plane. The same thresholds have been used for all images in the dataset. The obtained results shown in Figure 6 indicate that the individual use of these features correlates moderately with the annotations made by the experts, with pix H-score showing the lowest correlation.

We have found that including the normalization factors improves this correlation significantly, allowing a subsequent classification of the images more in line with the experts’ notation. Figure 7 shows a scatter plot of the features calculated as Equations (Equation 12), (Equation 14) and (Equation 15) and a reference truth calculated as the median between the scores of observers #1, #2, and #3. A higher correlation can be seen, although even with these new features, there are no clear boundaries to differentiate the five image classes.

The correlations between the scores provided by the proposed method and experts’ scores are summarized in Table 1 in terms of accuracy. A high level of accuracy is obtained in all cases. We have also investigated the inter-observer reliability for the dataset by computing the Cohen’s kappa values κ between pairwise comparison of the observers and the proposed method. The strength of agreement has been estimated using the rating scale proposed in [40] as follows: values κ<0.20 as indicating bad; 0.21–0.40 as fair; 0.41–0.60 as moderate; 0.61–0.80 as good; and 0.81–1.00 as very good agreement. Results are shown in Table 2. Although Cohen’s kappa values were variable between observers, the most common agreement was very good.

The method has been implemented in MatLab R2021b. The total execution time, including the stain separation, feature extraction, and score prediction of the 94 images was 244.48 s in an iMac (chip: Apple M1, RAM: 8GB).

## 5. Conclusions

In biomedical research, semi-quantitative scoring systems are widely used to convert the subjective perception of IHC marker expression by researchers into quantitative data that allow determining specific differences in a group of images. Manual scoring is not only time consuming but also suffers from the classical inter-observer, intra-observer, and inter-session variabilities. In addition, human physiological factors such as fatigue and eyestrain can occur during the monotonous process of evaluating a large number of images. Therefore, subjectivity is a mayor problem that can be avoided with an automatic system.

In this article, we have developed an automatic scoring system in order to emulate the scores that a researcher may propose. Our method consists of three parts: stain separation, feature extraction, and prediction of the scores. The staining separation step has been specifically designed to perform robustly on IHC images expressing the protein of interest at both very high and very low intensities. In addition, it is capable of automatically adjusting the color vectors in each image without the need for researcher intervention. With a proper separation of the color planes, it is possible to calculate different types of features, the simplest being pixel-wise features. However, we have found that the individual use of pixel-wise features correlates only moderately with the annotations provided by the experts.

The measures we propose as features for the prediction of the scores are related to the intensity of DAB staining and to the relative intensity DAB and H staining between the image and a control image. Relative intensity features are related to normalization factors of color appearance in histological images. These features significantly improve the correlation of the predicted scores with those provided by experts.This may be due to the fact that the perception of intensity can be altered for a variety of reasons. In particular, pale browns tend to be perceived less intensely than dark browns. In addition, the hue and saturation of colors may affect the perception of the intensity. The values of relative intensity features do not depend on the color bases determined at the stain separation stage, but it helps to reproduce the comparative intensity perception of the experts. The prediction of the scores is performed by an unsupervised clustering method in order to add some flexibility to the proposed method rather than setting predefined boundaries between the different classes.

The results obtained on a database of 94 images show that the developed method achieves its goal, predicting image scores automatically and highly correlated with the scores predicted by the experts. The scoring method reaches 94.58% in terms of accuracy with respect to one of the experts. The lower accuracy obtained has been 76.60%. These values can be considered as very good, mainly because we have used a five-point scale, which is a high number of possible categories in semi-quantitative studies. Similar results have been obtained with Cohen’s kappa coefficient, which is a statistical measure that adjusts for the effect of chance on the proportion of observed agreement. In particular, the strength of the agreement between the proposed method and the observers was very good except for one of the observers in which the strength was lowered to good. The proposed method can easily be adapted to a number of different categories, and it is able to handle with large datasets.

In the future, we would like to further investigate other features related to color perception that may model the influence of differences in true colors of the images on the scores provided by experts. In our opinion, the relevance of features related with color perception has been understated by the scientific community with current methods focusing on intensity-based features.

## Figures and Tables

**Figure 1 entropy-24-00546-f001:**
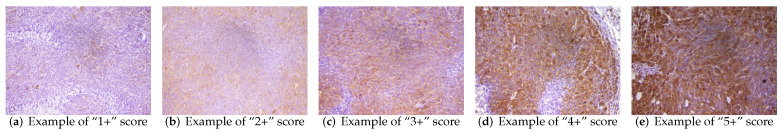
Examples of different immunolabeling intensities in IHC images with a magnification of 20×: (**a**) very low positivity or negative (1+), (**b**) low positivity (2+), (**c**) mild positivity (3+), (**d**) moderate positivity (4+), and (**e**) strong positivity (5+). The protein was visualized by DAB chromogen and nuclear counterstain with hematoxylin.

**Figure 2 entropy-24-00546-f002:**
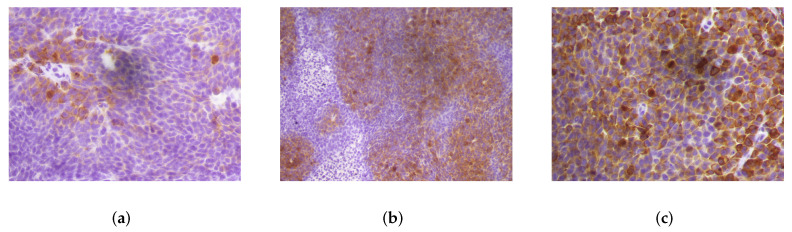
Example of IHC images where there is a discrepancy in the score assigned by the observers. In (**a**), the image has been annotated as 1+ or 2+, in (**b**), the image has been annotated as 3+ or 4+, and in (**c**) the image has been annotated as 4+ or 5+.

**Figure 3 entropy-24-00546-f003:**
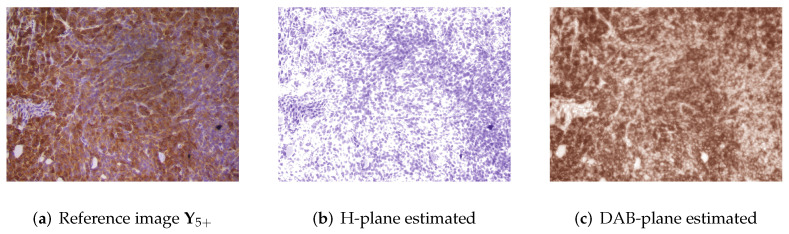
Stain separation results of the reference image $Y_{5+}$ obtained with eigendecomposition method: (**a**) Original IHC image (**b**) H-plane estimated (**c**) DAB-plane estimated.

**Figure 4 entropy-24-00546-f004:**
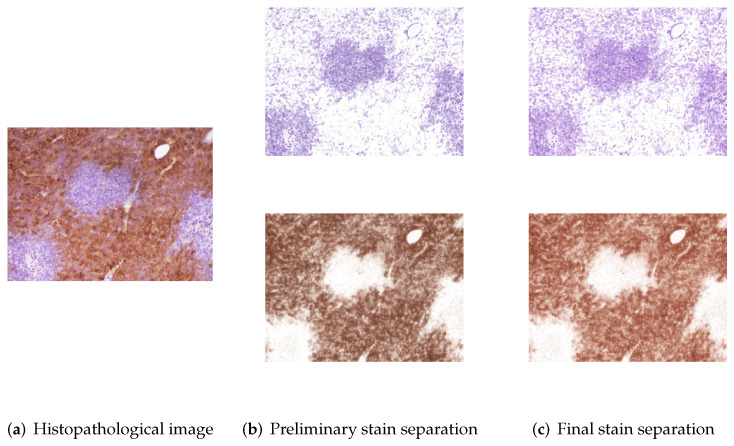
Example of stain separation in an IHC image with a high-intensity level: (**a**) Original image, (**b**) Preliminary stain separation and (**c**) Final stain separation.

**Figure 5 entropy-24-00546-f005:**
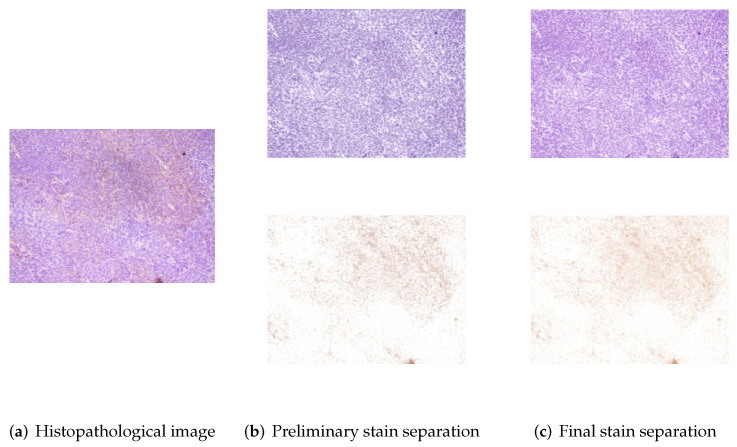
Example of stain separation in an IHC image with a low-intensity level: (**a**) Original image, (**b**) Preliminary stain separation, and (**c**) Final stain separation.

**Figure 6 entropy-24-00546-f006:**
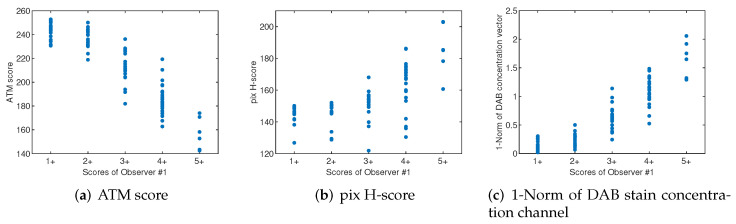
Correlation between some features based on the DAB staining plane and the score annotated by one expert for all the images in the dataset. The features examined are: (**a**) ATM score, (**b**) Pix-H score, and (**c**) 1-norm of the DAB stain concentration vector obtained in the NMF decomposition of the OD image. Similar results are obtained with the scores of the other observers.

**Figure 7 entropy-24-00546-f007:**
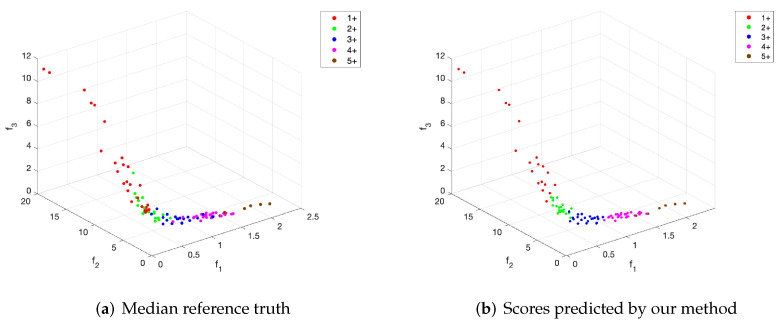
Scatter plot of the extracted features and the scores: (**a**) calculated as the median value of the annotations of observers, (**b**) predicted by our method.

**Table 1 entropy-24-00546-t001:** Comparison of the performance of different clustering algorithms in terms of accuracy (%).

	K-Means with Euclidean Distance	K-Means with Beta Divergence
Observer #1	93.61	94.58
Observer #2	75.53	76.60
Observer #3	86.17	87.23
Observer #4	89.36	90.43
Mean	86.17	87.23

**Table 2 entropy-24-00546-t002:** Pairwise inter-observer reliability of semi-quantitative scoring by four observers and the proposed score. Crosstabs contain the Cohen’s kappa values, κ (orange background), and the strength of agreement (blue background) between two different observers.

Observers	Observer #1	Observer #2	Observer #3	Observer #4	Predicted
Observer #1		0.7672	0.8503	0.8906	0.9315
Observer #2	Good		0.7810	0.6850	0.6979
Observer #3	Very good	Good		0.7054	0.8364
Observer #4	Very good	Good	Good		0.8765
Predicted	Very good	Good	Very good	Very good	

## Data Availability

The data that support this study are available from the author R.V.D., upon reasonable request.

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
