# Peer review of "A Method for Unsupervised Semi-Quantification of Inmunohistochemical Staining with Beta Divergences"

_entropy, 2022, doi:10.3390/e24040546_

Round 1
Reviewer 1 Report
The authors present a method for the automatic semi-quantification of images of tissue with immunohistochemical stainings. The paper focuses on the case of a DAB and hematoxylin double-staining as is used in cancer assays in biomedical research. The aim is to automatically classify the DAB staining in the images into 5 categories from very low to very high. The method presented starts by transforming the images into optical density space to take care of the non-linearity between the intensity and the stain concentration. A negative and a positive control image with respect to the DAB staining must be selected manually from the dataset. An initial color deconvolution is calculated based on the positive control image. The color deconvolution separates the RGB vectors of the N pixels of each image into a 3x2 color vector matrix and a 2xN stain density matrix. To calculate the initial color vector matrix, the eigendecomposition of the autocorrelation matrix of the positive control image is computed, the two eigenvectors corresponding to the largest eigenvalues are found and the corresponding coordinates are estimated. The initial stain densities for all images are then calculated using color deconvolution. The initial separation is adjusted for each image, using non-negative matrix factorization (NMF) with beta-divergence as approximation measure. 0.5 is chosen as a value for beta. The resulting stain density matrices are used to extract 3 features from each image, the 1-norm normalized to the image size of the DAB stain densities, the ratio of the robust pseudo maximum (RM) of the DAB channel of the control image to the RM of the DAB channel of the image and the second feature divided by the corresponding ratio in the hematoxylin channel. To classify the images into the 5 categories, a k-means clustering with beta-divergence is used. Again 0.5 is chosen for beta. Two of the initial means are set to features of the negative and positive control. The others are calculated as a percentage of the difference between the first feature of the two control images. The percentages have to be fixed for each experiment. The results of the initial and the adjusted color deconvolution are presented visually with the conclusion that the adjustment gives generally better results and that the separation also works well for the negative control image. The correlation between the first feature used in the proposed method and the classification of one observer is compared to the correlations obtained with the ATM-score and the pix H score, two features used in other methods, with the result that the proposed feature correlates better with the classification of the observer. Scatter plots of the 3 used features show that even with these features there are no clear boundaries between the classes, justifying the use of the k-means algorithm. The correlation of the predicted classes and the classifications of four observers is shown for the proposed method using the k-means with Euclidean distance and with beta-divergence. The correlation is generally good and slightly better in all cases for the beta-divergence version. The inter-observer reliability of the classification is shown for the four observers and the results of the method, using Cohen’s kappa values. The reliability of the method very good with respect to 3 of the observers and good with respect to one observer. The authors conclude that the method can successfully emulate the classification of researchers avoiding bias and tedious manual classification. The stain adjustment solves the problem of stain separation for very high and very low values. The proper separation of stains allows the extraction of different types of features and the use of the proposed relative intensity features significantly improves the correlation between predicted scores with those provided by the experts.
To have a robust automated method for semi-quantification of immunohistochemical stainings is of high importance in biomedical research, not only to overcome observer bias, but also to deal with large quantities of data which can easily by acquired with modern acquisition technologies, making the image analysis the bottleneck. The presented study is therefore highly relevant. The main originality of the method consists in first automatically calculating an initial color deconvolution on the positive control image, without a manual choice of color vectors and then refining the results on a per image basis using non negative matrix factorization. Thus solving the problem present in other methods of achieving good results for images with very high and very low staining density in the same time. The usage of a non symmetric divergence measure is another interesting point that deserves further exploration. It would have been interesting to compare the results of the method with other more complex methods than those chosen, however the main point that the method can emulate manual scoring by experts has successfully been demonstrated.
The article is generally well structured and the argumentation well presented. I therefore only have some minor remarks, concerning some specific points that might be confusing for the reader.
In the introduction the authors correctly note that DAB does not only absorb but also scatter light and that for this and other reasons that can be found in citation [1] it does not follow the Beer-Lambert law. In the Related work section it is mentioned, that: "Stain separation is usually addressed in optical density (OD) space where the intensity can be assumed to be linear with the amount of each stain absorbed by a tissue sample.". To avoid confusion, it should be clearly stated that this does not hold for DAB stainings. Which could also be the reason for the non-linear relationship of the features in "prediction of scores' section.
It should be clarified that the DAB and H channels mentioned in the description of the ATM score and the pix H-score are not the stain densities maps, for which the same names are used in other places of the article. For the ATM-score the "DAB channel" is obtained as the complement of the blue channel in the original RGB image.
Section 3.1.1 starts with mentioning the positive and negative control images. I therefore expected that these would be used afterwards. However only the positive control is specifically used in the whole stain separation process.
If I'm not missing something evident, there seems to be a problem with the indices in the multiplicative update rule (11), the index of the sum in the denominator i collides with the outer index i.
I wonder how the choice of the value 0.5 in the beta divergence is motivated. Do the authors have a special reason to choose this value or is it just to explore other dissimilarities measures?
In section 4.3 the authors say they have used the median between the scores of observers #1, #2 and #3 as a reference truth. It is not clear to me where and how this reference truth has been used.
The yellow dots used for class 5+ in figure 7 are barely visible on my screen. It could be advisable to use a darker color, for example brown, especially since it would be interesting to see how well the class 5+ dots are separated from the class 4+ dots.
Author Response
We thank the reviewer #1 for taking the time to review our manuscript and for the constructive and insightful comments, which have helped us to substantially improve our manuscript.
We have addressed all the comments in the revised manuscript as discussed below. Throughout, reviewer comments are in black regular type, and our response in red regular type. There have been textual changes throughout the manuscript; the most significant additions and rewrites have been highlighted in red (related to comments of reviewer #1), and blue (related to comments of reviewer #2).
Our manuscript has been reviewed by a colleague and revised to improve readability.

Reviewer 2 Report
Authors introduce new approach in scoring of immunohistochemical images using first deconvolution and then k-mean clustering.
The paper is presented well, but the novelty of the paper is limited and can be improved. Also, background review and comparison with the state-of-art approaches should be improved to clarify the novelty of the proposed technique.
The implementation of the method is another subject that is missing in the article. Which programming language is employed to implement the idea? How fast does it provide the outcome (scores)? How does it perform in comparison to others? Are the code and datasets shared with public (i.e. the proposed method is reproducible?)
Author Response
We thank the reviewer #2 for taking the time to review our manuscript and for the constructive and insightful comments, which have helped us to substantially improve our manuscript.
We have addressed all the comments in the revised manuscript as discussed below. Throughout, reviewer comments are in black regular type, and our response in blue regular type. There have been textual changes throughout the manuscript; the most significant additions and rewrites have been highlighted in red (related to comments of reviewer #1), and blue (related to comments of reviewer #2).
Our manuscript has been reviewed by a colleague and revised to improve readability.
